# Co-Expression Network Analysis of Spleen Transcriptome in Rock Bream (*Oplegnathus fasciatus*) Naturally Infected with Rock Bream Iridovirus (RBIV)

**DOI:** 10.3390/ijms21051707

**Published:** 2020-03-02

**Authors:** Ahran Kim, Dahye Yoon, Yunjin Lim, Heyong Jin Roh, Suhkmann Kim, Chan-Il Park, Heui-Soo Kim, Hee-Jae Cha, Yung Hyun Choi, Do-Hyung Kim

**Affiliations:** 1Department of Chemistry, Center for Proteome Biophysics, and Chemistry Institute for Functional Materials, Pusan National University, Busan 46241, Korea; ahran110@naver.com (A.K.); dahyeyoon@korea.kr (D.Y.); suhkmann@pusan.ac.kr (S.K.); 2Department of Aquatic Life Medicine, College of Fisheries Science, Pukyong National University, Busan 48513, Korea; dladbswls@korea.kr (Y.L.); tlsrma92@daum.net (H.J.R.); 3Department of Herbal Crop Research, National Institute of Horticultural and Herbal Science, RDA, Eumseong 27709, Korea; 4Hazardous Substances Analysis Division, Gwangju Regional Office of Food and Drug Safety, Gwangju 61012, Korea; 5Department of Marine Biology and Aquaculture, College of Marine Science, Gyeongsang National University, Tongyeong 53064, Korea; vinus96@daum.net; 6Department of Biological Sciences, College of Natural Sciences, Pusan National University, Busan 46241, Korea; khs307@pusan.ac.kr; 7Department of Parasitology and Genetics, Kosin University College of Medicine, Busan 49267, Korea; hcha@kosin.ac.kr; 8Department of Biochemistry, College of Oriental Medicine, Dongeui University, Busan 47227, Korea; choiyh@deu.ac.kr

**Keywords:** Rock bream iridovirus, RBIV, co-expression network, WGCNA, transcriptome, integration, metabolome, pre-mRNA processing factor 19 (PRPF19)

## Abstract

Rock bream iridovirus (RBIV) is a notorious agent that causes high mortality in aquaculture of rock bream (*Oplegnathus fasciatus*). Despite severity of this virus, no transcriptomic studies on RBIV-infected rock bream that can provide fundamental information on protective mechanism against the virus have been reported so far. This study aimed to investigate physiological mechanisms between host and RBIV through transcriptomic changes in the spleen based on RNA-seq. Depending on infection intensity and sampling time point, fish were divided into five groups: uninfected healthy fish at week 0 as control (0C), heavy infected fish at week 0 (0H), heavy mixed RBIV and bacterial infected fish at week 0 (0MH), uninfected healthy fish at week 3 (3C), and light infected fish at week 3 (3L). We explored clusters from 35,861 genes with Fragments Per Kilo-base of exon per Million mapped fragments (FPKM) values of 0.01 or more through signed co-expression network analysis using WGCNA package. Nine of 22 modules were highly correlated with viral infection (|gene significance (GS) vs. module membership (MM) |> 0.5, *p*-value < 0.05). Expression patterns in selected modules were divided into two: heavy infected (0H and 0MH) and control and light-infected groups (0C, 3C, and 3L). In functional analysis, genes in two positive modules (5448 unigenes) were enriched in cell cycle, DNA replication, transcription, and translation, and increased glycolysis activity. Seven negative modules (3517 unigenes) built in this study showed significant decreases in the expression of genes in lymphocyte-mediated immune system, antigen presentation, and platelet activation, whereas there was significant increased expression of endogenous apoptosis-related genes. These changes lead to RBIV proliferation and failure of host defense, and suggests the importance of blood cells such as thrombocytes and B cells in rock bream in RBIV infection. Interestingly, a hub gene, pre-mRNA processing factor 19 (PRPF19) showing high connectivity (kME), and expression of this gene using qRT-PCR was increased in rock bream blood cells shortly after RBIV was added. It might be a potential biomarker for diagnosis and vaccine studies in rock bream against RBIV. This transcriptome approach and our findings provide new insight into the understanding of global rock bream-RBIV interactions including immune and pathogenesis mechanisms.

## 1. Introduction

Iridoviridae is a family of large cytoplasmic double-stranded DNA virus consisting of five genera (Iridovirus, Chloriridovirus, Ranavirus, Lymphocystivirus, and Megalocytivirus) with icosahedral morphology. These viruses can infect insects, amphibians, and fish. Rock bream iridovirus (RBIV) is cladded with red sea bream iridovirus (RSIV), a subgroup of genus Megalocytivirus [1]. It causes significant mortality of rock bream (*Oplegnathus fasciatus*) [2,3,4]. It has been reported that RSIV can infect more than 30 other species of farmed marine fish mainly belonging to orders Perciformes and Pleuronectiformes [5]. Although a formalin-killed vaccine for RSIVD is commercially available [6], it is not so effective in protecting rock bream [4]. 

In recent years, genome, metagenome, and transcriptome analyses such as case-control studies of diseases have been actively conducted for various organisms ranging from microorganisms to animals and plants due to advances in next-generation sequencing (NGS) technology [7,8,9,10]. While whole-genome analysis offers genetic and regulatory information only, transcriptome analysis based on RNA-seq not only allows quantification of gene expression profiles for various conditions, but also provides new transcript information such as nucleotide variations and alternative splice variants [11]. In addition, it is possible to analyze at a higher sensitivity and wide dynamic range without prior knowledge of the reference than sequence-based transcript profiling such as microarray [11,12,13]. Transcriptome analyses based on RNA-seq have led to the discovery of novel genes of host, providing a better understanding of pathogenic processes and immune responses through qualitative and quantitative analyses and functional annotations during diseases caused by pathogens. Results of such analyses also provide foundation for control strategies against diseases [12,13].

Many studies have been reported on clinical signs of rock bream infected by RBIV [14], RBIV complete genome [3,15], RBIV transcriptional profiles [15], expression of apoptotic proteins coding genes [16,17], proinflammatory genes [18], or several immune-related genes [19,20]. Despite these efforts, little is known about how rock bream defends itself against virus and how RBIV invades into rock bream and overcomes immune responses of the host. RNA-seq has been performed for spleen of Chinese giant salamander (*Andrias davidianus*) infected with GSIV [21] and orange-spotted grouper (*Epinephulus coioides*) infected with Taiwan grouper iridovirus (TGIV) or grouper iridovirus (GIV) through *de novo* assembly and differentially expressed genes (DEGs) approach [22,23]. However, transcriptome analysis of rock bream infected with RBIV has never been reported. Additionally, it is still difficult to systematically understand biological changes in the host caused by various factors due to insufficient database and information for fish and the lack of understanding in the methodology for transcriptome analysis.

Genes play a role by interacting with each other rather than by acting alone [24,25]. Functions of many genes are still poorly understood. Their interactions are also very complicated to understand [24]. A construction of gene co-expression network is effective in characterizing correlation patterns between genes [25]. A gene-gene co-expression analysis has emerged as a powerful approach to solve these problems described above. This approach is based on clustering by global similarity in gene expression profiles [26,27]. Genes showing similar expression patterns are closely related. Their biological functions might be similar or regulated by each other [28,29]. This co-expression analysis could provide better understanding of gene function, gene-disease association, and disease origin and progression. It has been used to identify functional gene annotation or regulatory genes [30,31,32]. WGCNA is a clustering algorithm that describes correlation patterns of gene transcripts. It presents a gene-gene similarity matrix with high-throughput data. It divides genes with similar expression pattern into gene modules [33].

In this study, full-length transcripts with isoform profiles of genome-free rock bream were established to construct more reliable reference transcriptome. RNA-seq of spleen, the main target organ of RBIV in rock bream, was performed through NGS approach. We conducted weighted gene correlation network analysis (WGCNA) for the expression profile of these transcript data and analyzed interactions between gene expression pattern and the degree of viral infection. One ultimate objective of this study was to investigate transcriptional changes occurring in rock bream after RBIV infection through functional analysis of genes in the modules and gene expression analysis by qRT-PCR. In addition, we tried to confirm changes in metabolites of spleen by NMR-based metabolomics integrated with WGCNA analysis results to understand biological functions involved in simultaneous transcript and metabolism. 

## 2. Results

### 2.1. Clinical Traits

Cumulative mortalities in three weeks were 92% and 0% for the heavily infected group and the negative to lightly infected group, respectively. Mortality in heavily infected group was 36.5, 55.5%, and 0% in a total population at week 1, 2, and 3, respectively. Average splenic viral load and spleen index (SI) of fish taken from both 0MH (mixed heavy RBIV and bacterial infected fish at week 0) and 0H (heavily RBIV-infected fish at week 0) groups were above 10^8^ copies/mg of spleen and over 2, respectively (Appendix A). These were significantly higher than those of fish in 0C (uninfected healthy fish at week 0) and 3C (uninfected healthy fish at week 3) groups. Splenic viral load of fish in the 0MH group (2.56 ± 1.31 × 10^8^ copies/mg of spleen) was two times higher than that in the 0H group. All five fish in the 0MH group were also infected with *Vibrio harveyi* (99.70% identity) and/or *V. scophthalmi* (99.66% identity) identified by 16S rRNA gene sequencing. Although fish (*n* = 16) in the 3L (light RBIV-infected fish at week 3) group showed no signs of RBIV infection, mean splenic viral load of five randomly selected fish was 4.15 ± 4.14 × 10^3^ copies/mg of spleen. 

### 2.2. Module Detection Highly Correlated with RBIV Infection by WGCNA

Principal component analysis (PCA) and clustering dendrogram with expression profile of 35,861 genes show clustering according to experimental groups except for one sample in 3C group (Appendix A). In particular, samples from 0MH and 0H groups were tightly clustered (Appendix A), indicating there were no significant differences in transcriptome expression patterns between these two groups. A total of 22 modules were created by WGCNA (Figure 1**A**), and they were labeled by colors as described in y-axis of Figure 1B. Module-trait correlations calculated with splenic viral copy numbers, SI, presence of bacterial infection, and relative concentration of metabolites are shown in Figure 1B. Nine modules showed significant correlations with splenic viral loads (|gene significance (GS) vs. module membership (MM)| > 0.5 and *p*-value ≤ 0.05) positively (5448 unigenes, called positive modules) or negatively (3517 unigenes, called negative modules) (Table 1 and Figure 2). Gene expression patterns of heavy infected fish groups (0MH and 0H) were confirmed to be up- and down-regulated in positive and negative modules, respectively (Figure 2B). In this study, thirty-three metabolites were identified by high-resolution magic angle spinning nuclear magnetic resonance (HR-MAS NMR) spectroscopy (shown in Appendix A).

### 2.3. Functional Enrichment Analysis of Modules

For functional annotation, 4813 (53.69%) and 5438 (60.66%) unigenes among 8965 unigenes were assigned in KEGG and GO, respectively. The most significant KEGG pathways (*q*-value < 0.05) and over-represented GO terms (adjusted *p*-value < 0.05) of nine modules are presented in Figure 3 and Figure 4, respectively. The most significant terms and pathways for each module are shown in Table 1. Results showed that specific functions varied depending on modules. Notably, in KEGG analysis, it could not confirm signaling and immune-related pathways in positive modules, and pathways belonging to genetic information processing and metabolism categories in negative modules. Annotated genes in interesting biological functions using KEGG Automatic Annotation Server (KAAS) and Biological Networks Gene Ontology (BiNGO) are shown in Appendix A.

#### 2.3.1. Up-Regulation of Genes Involved in Proliferation of RBIV in Spleen

(1)Cell Cycle, DNA Replication, and Cell Proliferation

DNA replication, DNA repair, and cell cycle were enriched in Turquoise. Especially, genes involved in mitosis were up-regulated in heavy-infected groups (Figure 3A and Figure 4A). Almost all genes involved in spindle checkpoints and G2/M transition DNA damage checkpoints in the cell cycle were found in Turquoise (Figure 4A): Cyclin A(*CCNA*)-cyclin-dependent kinase (*CDK1*), cyclin B (*CCNB*)*-CDK1* complex, *CDC25* phosphatase, M-phase inducer phosphatase, mini-chromosome maintenance complex (*MCM*, Turquoise), checkpoint kinase 1 (*Chk1*), and Wee1-like protein kinase (*WEE1*) (Appendix A).

(2)Transcription and Translation

Pathways belonging to transcription and translation categories were highly significant in Turquoise, especially spliceosome and RNA transport (Figure 3A and Figure 4A). In spliceosome, 78 of 105 unigenes were up-regulated in heavy-infected groups (Appendix A). In RNA transport pathway, 122 of the total 131 unigenes were in Turquoise, especially various components required for different RNA species transport including eukaryotic initiation factors (*eIF*s), nuclear pore complex (NPC; *Nups*), GTP-binding nuclear protein *Ran*, exportin, importin, and exon-junction complex (EJC) were identified in Turquoise. All 46 genes mapped on aminoacyl-tRNA biosynthesis pathway in KEGG were identified in Turquoise (Appendix A). KEGG enrichment analysis results (in Supplementary Methods and Results) of 33 metabolites obtained by NMR-based metabolomics are shown in Appendix A. Through integrated analysis with transcripts in each gene cluster, the aminoacyl-tRNA biosynthesis pathway was mapped with 13 metabolites in Turquoise (Appendix A).

(3)Protein Processing in Endoplasmic Reticulum (ER)

Protein processing in endoplasmic reticulum (ER) was potently enriched in Turquoise (110 of 127 unigenes in total). It showed that genes involved in endoplasmic reticulum and Golgi apparatus (G)-related transport, especially by coat protein complex (COPII)-c oated vesicle from ER to G (11 unigenes), ER stress and unfolded protein response (UPR) (8 unigenes), and ER-associated degradation (ERAD) (19 unigenes) were up-regulated in heavy-infected groups (Figure 3A and Figure 4A, and Appendix A).

(4)Metabolism

In carbohydrate metabolism category, citrate cycle (TCA cycle) and glycolysis/gluconeogenesis pathways were significant in Turquoise. All 31 genes in TCA cycle were up-regulated (Figure 3A, Appendix A). In addition, expression levels of genes involved in glycolysis including *HK1, PFKL, PFKP,* and *LDHA* were significantly up-regulated in heavy-infected groups (Appendix A).

(5)Apoptosis

Among a total of 65 genes in apoptosis pathway, 44 genes were up-regulated. Notably, *CASP3*, *CASP6*, *CASP9*, and *CYC* were up-regulated, whereas *BCL2L1* was down-regulated significantly in heavy-infected groups (Appendix A). 

#### 2.3.2. Host Immune Defense Failure against Virus Infection

(1)Decreased Platelet Activation

Platelet activation (13 unigenes), platelet α-granule (4 unigenes), and response to wounding (3 unigenes) were enriched in Magenta (Table 1, Figure 3B and Figure 4B). Of the total 60 unigenes annotated in the platelet activation pathway, 32 unigenes were significantly down-regulated in heavy-infected groups. In particular, expression levels of fibrinogen gamma chain (*FGG*, Blue), glycoprotein (GP) IIb/IIIa (integrin *αII*b*β3* or CD41/61, Magenta), and GPIb/IX/V complex (CD42b/42a/V; Magenta/Blue/Magenta) were decreased (Appendix A).

(2)Immune Systems

In KEGG analysis, immune-relevant pathways were observed only in negative modules (Figure 3B). In pattern recognition receptors pathways, *TLR7* and *TLR9* (both Purple) were significantly down-regulated in heavy-infected groups (Appendix A). In Magenta, C1 complex was enriched (Figure 4B) while genes associated with complement and coagulation cascade including *C1qa*, *C1qb*, and *C1qc* were significantly down-regulated (Appendix A). MHC class I (Grey, not selected modules) and II (Red, Purple, and Brown) were down-regulated in heavy-infected groups. In particular, B cell signaling pathway (in Purple) was the most potently enriched among negative modules while *CD79* (Igα and β; Purple), *CD22* (Purple, Red), *BLNK* (Red), *SYK* (Purple), and *BTK* (Purple) were down-regulated in heavy-infected groups. In addition, significant changes were identified in heavy-infected groups, including up-regulated cytokines (*IL1β* and IFN-γ), down-regulated chemokines (*CXCL12* and *CXCL14*), down-regulated their receptors (CXCR and CCR), and up-regulated other immune-related genes such as *IRF4* and up *CTLA4* (Appendix A). 

(3)Disrupted Cytoskeleton and Cell-To-cell Interaction

Cytoskeleton- and cellular community-associated terms and pathways appeared in negative modules (Table 1, Figure 3B and Figure 4B). Spectrin binding (in Blue) was enriched while *STPA*, *STPB*, *ANK1*, and *EPB41* were down-regulated significantly. Almost all genes, especially collagen, laminin, and integrin in focal adhesion, adherens junction, and ECM-receptor interaction pathways annotated in KEGG, were down-regulated in heavy-infected groups (Appendix A).

(4)Signaling Pathways

Signaling pathways were found in negative modules. Genes in Wnt signaling pathway (Green), Rap1 signaling pathway (Green, Magenta, Pink), PI3K-Akt signaling pathway (Green, Magenta), and IKK/NF-kappa B cascade (Purple) were significantly down-regulated (Figure 3B and Figure 4B).

### 2.4. Hub Genes Analysis in Selected Modules

We investigated enriched functions of top 30 hub genes to determine how they regulated and interacted with neighboring genes in each module. Results were similar to those of all genes in each module, suggesting that these top 30 hub genes might perform representative functions. They might be strongly involved in each module (Appendix A). Among the top 5 hub genes (Appendix A), pre-mRNA processing factor 19 (*PRPF19*) with high connectivity (kME) and correlation with infection in Turquoise (the largest module in this study) was selected as a possible genetic marker. It was used in the following gene expression analysis.

### 2.5. Gene Expression in Rock Bream Blood Cells Infected with RBIV

We identified down-regulation of genes involved in platelet activation, B cell receptor signaling pathway, and spectrin-associated cytoskeleton in our transcriptome results. Based on this, we hypothesized that RBIV would interact with various types of blood cells including thrombocytes, erythrocytes, and lymphocytes such as B cells. The identified genes were selected from enriched biological functions in our transcriptomic results: *TLR9* in pattern recognition receptor pathway, *CASP3* in apoptosis, *H2L* and *MHCIIa* in antigen processing and presentation pathway, *CTLA4* in immune response, *ITGA2B* and *GP5* of platelet activation pathway (Appendix A). Thus, expression of genes in blood cells with plasma removal from rock bream was examined after incubating with RBIV. After inoculation of RBIV, expression levels of *TLR9* and *CASP3* were increased significantly at 120 hpi and 6 hpi, respectively (Figure 5A,B). *H2L* was significantly up-regulated at 12 and 120 hpi (Figure 5C), whereas MHC IIα showed a slight but not significant down-regulation up to 120 hpi (Figure 5D). Notably, gene expression patterns of *H2L* and *TLR9* were contrary to our transcriptome results. *CTLA4* expression increased immediately after inoculation. It was significantly increased at 120 hpi (Figure 5E). Both ITGA2B and GP5 were down-regulated significantly at 3 and 24 hpi (Figure 5G,H). *PRPF19* was increased from soon after viral inoculation up to 120 hpi, similar to the increase of virus recognition (*TLR9*) or immune-relevant gene expression (*CTLA4*) in blood cells infected with RSIV. Expression levels of *PRPF19* were 2.77-fold and 9.00-fold higher at 6 hpi and 120 hpi, respectively (Figure 5F).

## 3. Discussion

In the present study, we performed reference transcriptome construction of genome-free rock bream using full-length transcripts sequencing with PacBio platform and obtained 68,211 unigenes from the liver, spleen, and head kidney (Appendix A). Approximately 84.75% of transcript data from the spleen of RBIV-infected or non-infected samples were mapped onto full-length transcripts (Appendix A). Whereas it was possible to analyze gene sequences and expression profiles by sequence-based analysis such as microarray- and tag-based approaches in a previous study [34], full-length transcript sequencing (Iso-Seq) could help us produce *de novo* full-length open reading frames (ORFs) and comprehensive transcriptome with accurate alternative spliced isoforms. This also allows for deeper genome-wide transcriptome analysis or full-length spliced isoforms analysis for various organs and organisms [35,36,37].

Expression patterns of nine gene clusters constructed using WGCNA in this study were divided largely into two: heavy-infected (0H and 0MH groups), and control and light-infected groups (0C, 3C, and 3L groups). PCA shows that two groups, 0H and 0MH, have very similar gene expression patterns, indicating that they were not affected by bacterial infection in fish of 0MH group. Perhaps, this was because secondary bacterial infection occurred after severe viral infection. In addition, the overall level of vibrio (*V. harveyi* and/or *V. scophtalmi*) infection was not severe as the number of bacterial colonies was low (approximately 3-30 per fish spleen). Although they were unbiased and scale-free analyzed through WGCNA without prior knowledge about function, genes in each module revealed similar and interconnected biological functions (Figure 3 and Figure 4, Table 1). It helped us analyze and understand biological pathways using not only significant genes statistically, but also insignificant yet meaningful genes. Through functional enrichment analysis in this study, blood cells including red blood cells, lymphocytes and thrombocytes of fish were severely affected by infection with RBIV. It is known that fish erythrocytes are involved in immune responses such as chemokine signaling pathway, platelet activation, and T cell receptor signaling pathway [38]. In proteome study of Jung et al. [39], apoptosis-, MHC I-, and spliceosome-related pathways in rock bream RBCs were up-regulated, and antiviral mechanisms were down-regulated in the response to RBIV infection. In recent years, more evidence has emerged that platelet and their activation state can modulate innate and adaptive immune responses [40]. Therefore, blood taken from RBIV-negative rock bream was used for the determination of RBIV infection in various types of cells in the blood on pathogen pattern recognition receptors, apoptosis, antigen processing and presentation, lymphocyte-mediated immune response, and platelet activation.

In our transcriptome results, pathways associated with DNA replication, cell cycle, RNA transport, transcription, translation, and cellular metabolism for virus proliferation were significantly enriched in positive modules. Although viruses have different strategies to replicate their genomes, viruses can replicate using host machinery such as controlling cell cycle checkpoints or cell proliferation pathways [41]. Viral proteins of human immunodeficiency virus type 1 (HIV-1) [42] and human papillomavirus (HPV) [43] can induce G2/M arrest in host cell cycle and make virus facilitate replication to the maximum level [44]. In regard to this, in the present study, genes in both DNA damage repair and cell cycle including mitosis and spindle checkpoints and G2/M transition DNA damage check points [45,46] appeared to be active in heavily infected fish (Appendix A), suggesting that host cell cycle might be used for virus replication even in fish cells with high copy numbers of RBIV. 

Human viruses such as human influenza virus [47] and Papillomavirus [48] can utilize RNA splicing to generate mature mRNA for translation and facilitate their gene expression. This splicing process is a post-transcriptional mechanism. It is performed by a large ribonucleoprotein (RNP) complex called spliceosome [47,49]. We also confirmed that expression levels of genes involved in transcription and translation were up-regulated in heavy-infected fish (Figure 3A, Appendix A). Similarly, transcriptome analysis of orange-spotted grouper (*Epinephelus coioides*) infected with grouper iridovirus of Taiwan (TGIV, in Megalocytivirus) has revealed that the infection is strongly associated with spliceosomal pathway [23].

In general, viruses can utilize ER functions to enhance their life cycle such as genome replication, protein folding, assembly, and egress through COPII vesicle [50,51]. Megalocytivirus is characterized by the presence of inclusion body-bearing cells containing fine and coarse granules and viral assembly site (VAS) with many viral particles in infected tissues [52]. It has been shown that replication and assembly of new virions are carried out at viral assembly site (VAS) in host cell infected with frog virus 3 (FV3) [53], Singapore grouper virus (SGIV) [54], or Chinese giant salamander iridovirus (GSIV) [55]. In addition, ER stress caused by accumulation of misfolded proteins such as virus-derived protein can lead to unfolded protein response (UPR) that can induce an inflammatory response or apoptosis in host, or ER-associated degradation (ERAD) that eventually degrades the misfolded substrate to the cytoplasm [51]. We observed that genes involved in ERAD, UPR, and COPI/COPII vesicle were significantly up-regulated in heavily infected fish. These results can be seen that activity in ER and G is increased during RBIV infection, although the exact mechanism has not been clarified yet.

Glycolysis in normal cells produces pyruvate when oxygen is sufficient. It efficiently produces a large amount of ATP through the TCA cycle and electron transport chain in mitochondria [56]. However, in some virus-infected and cancer cells, aerobic glycolysis called Warburg effect can occur in the cytoplasm, resulting in increased conversion of pyruvate into lactate even when oxygen is present [57]. These cells are characterized by increased glucose consumption, lactate production, and reduced oxygen consumption [56]. Although oxidative phosphorylation provides much more ATP per glucose molecule, glycolysis provides ATP much faster in cells [58,59]. Both TCA cycle and glycolysis/gluconeogenesis pathway were significant in Turquoise in this study to generate energy actively (Figure 3A and Figure 4A, Appendix A). Significant up-regulation of genes in glycolysis/gluconeogenesis pathway was also found in orange-spotted grouper infected with GIV [23]. DEGs mapped to this pathway were mostly consistent with our data. Although NMR-based metabolomics profiling in the present study revealed no significant difference in glucose level between groups, lactate levels were increased in the heavily infected group, especially in 0MH group (Appendix A). Lactate also had the highest VIP score (Appendix A), indicating that it showed the greatest difference between heavy infection groups (0H and 0MH) and other groups (0C, 3C, and 3L). KEGG pathway analysis of metabolites also indicated that central carbon metabolism in cancer pathway was significantly up-regulated. These results support that the Warburg effect might occur in RBIV infected fish.

Our RNA-seq data showed that expression levels of genes involved in intrinsic apoptosis pathway mediated by mitochondria (Appendix A) were significantly increased. In GF cells infected with RSIV, caspase-dependent apoptosis was induced during permissive replication. Caspase-3 and -6 are involved in morphological changes [60]. It has been reported that caspase-3 is significantly up-regulated in the late phase of RBIV infection in the liver of rock bream [16]. Apoptosis-related proteins including *CASP6, CASP9*, and *Fas* in red blood cells were up-regulated in proteomic profiling [39]. Expression level of *CASP3* in rock bream blood cells infected with RBIV is increased significantly at 6 hpi (Figure 5B). Taken together, these results indicate that RBIV could induce apoptosis in blood cells and splenic cells of rock bream. 

Pathways associated with signaling pathways and host defense such as lymphocyte-mediated immune system, platelet activation, cytoskeleton-related pathways were significantly enriched in negative modules. In this study, platelet activation and response to wounding were significant pathways in Magenta (Figure 3B and Figure 4B). Fibrinogen gamma chain (FGG), glycoprotein (GP) IIb/IIIa (integrin αIIbβ3 or CD41/61) and GPIb/IX/V complex (CD42b/42a/V) as a receptor for von Willebrand factor (vWF) and Mac-1 were down-regulated in heavily infected fish (Appendix A). In humans, dysfunctions of those might lead to no fibrinogen bridging between platelets [61]. Thus, platelets cannot be adhered to damaged blood vessel walls, leading to prolonged bleeding time [62]. Although RbCD42c (GPIbβ) was highly expressed in red blood cells of healthy rock bream, this gene was significantly down-regulated in the kidney, spleen, liver, and gill at 24 h after infection with RSIV [63]. In fact, thrombocyte in lower vertebrate including birds, amphibian and fish are a very specialized effector cell for hemostasis similar to mammalian platelet [64]. In our study, *ITGA2B* and *GP5* in blood cells incubated with RBIV were significantly down-regulated at 3 and 24 hpi, respectively (Figure 5G,H). Although this in vitro study is not enough to confirm occurrence of thrombocytopenia, the result suggests that heavy infection with RBIV might be associated with the disorder caused by increase in destructive thrombocyte and down-regulation of GPIIb/IIIa and GPIb/IX/V complexes expression on thrombocyte membrane. Formation of bridge between platelets or binding with fibrinogen and binding to vWF in blood vessel might be interfered by RBIV infection. These interferences might cause anemia due to increased bleeding without coagulation in heavily infected fish. Furthermore, gene expression in Magenta was up-regulated in light-infected fish (3L) compared to other groups in this study. It implies that thrombocyte activity in rock bream plays a key role in the progression of RBIV infection.

In our results, autophagy was found to be enriched in Blue and Brown. It could induce innate immunity and inflammatory responses in association with pattern recognition receptors (PRRs) and degradation of viral particles in antigen-presenting cells (APCs). This viral-derived antigen is presented to T lymphocytes using major histocompatibility complex (MHC) molecules [65]. From RNA-seq data in this study, endosomal TLR7 and TLR9 (both Purple) that could detect single-stranded RNA and DNA with unmethylated CpG sites, respectively, were significantly down-regulated in heavily infected fish. In addition, most genes involved in antigen processing and presentation pathway, especially MHC class I and II, in KEGG were down-regulated in heavily infected fish. However, expression levels of *TLR9* (2655-fold at 120 hpi) and MHC Iα (*H2L*) in blood cells infected with RBIV were higher than those in the control group (Figure 5A and 5C). The authors of a previous study [60] have also confirmed the up-regulation of antigen processing and presentation through the MHC class I pathway in proteomic profiling of RBCs in rock bream infected with RBIV. This suggests that RBIV is recognized by TLR9 to induce MHC class I expression for antigen processing and presentation during the early stage of infection, although RBIV might downregulate MHC-I and -II expression on RBIV-infected cells during the late phase of RBIV infection in rock bream. This is consistent with previous data demonstrating that influenza A and B viruses can downregulate MHC class I expression on IAV/IBV-infected cells in vitro [66].

Since host signaling pathways play important roles in many cellular processes, viruses target these pathways in various ways [67]. For example, SGIV can activate ERK signaling in *Epinephelus akaara* grouper spleen (EAGS) cells, resulting in induction of SGIV replication and nonapoptotic cell death [68]. LMBV infection can lead to virus production and induction of apoptosis by PI3K and ERK signaling pathways [69]. In the present study, Rap1 signaling pathway was significant in Magenta, Pink, and Green. This pathway is involved in functions of hematopoietic cells including platelet functions, megakaryocyte maturation, leukocyte adhesion, and cell growth [70]. In addition, lymphocyte activation, regulation, and differentiation were significantly enriched in Purple (Figure 3B and Figure 4B). B cell receptor signaling pathway essential for development and differentiation of B cells was potently enriched in Purple. These results suggest that the regulation of these pathways by RBIV infection might have inhibited or down-regulated transcription activity associated with B cells’ functions. Interestingly, cytotoxic T lymphocyte-associated protein 4 (*CTLA4*), a protein receptor expressed in regulatory T cell known to downregulate immune responses [70,71], was up-regulated in heavily infected fish (Appendix A) and blood cells (Figure 5E).

Many viruses can efficiently utilize cell adhesion mechanism to enter and spread to cells through direct cell-cell contact or binding with proteins [72]. In megalocytivirus, microfilament plays an important role in replication cycle of Infectious spleen and kidney necrosis virus (ISKNV) in MFF-1 cells [73] and egress for newly synthesized virus in grouper embryonic cell infected with SGIV [53]. Spectrin-ankyrin binding-related terms in Blue was down-regulated in the present study (Figure 4B). Spectrin is a cytoskeletal protein involved in cell adhesion, spreading, cell cycle, and intracellular traffic. It also maintains the mechanical stability of erythrocyte and non-erythrocytic cells [74,75]. In particular, protein 4.1 (*EPB41*) is a major component of the erythrocyte membrane skeleton that helps stabilize spectrin-actin interactions [75,76]. Protein 4.1R-deficient zebrafish showed red cell membrane disorder, severe hemolysis, cardiomegaly, and splenomegaly [76]. Most genes involved in cytoskeleton and cellular community pathways crucial for cell movement were down-regulated in heavily infected fish. These results suggest that as RBIV-infection progresses, down-regulation of cytoskeleton and cell adhesion in fish might inhibit cell functions such as maintenance of cell shape and integrity, cell attachment, spread, and signal transduction.

A hub gene is a regulatory gene that can have a major impact on a genetic network and affect the trait of interest. Highly interconnected hub genes from each module were identified (Appendix A), and they play important roles in biological processes (Appendix A). Interestingly, pre-mRNA processing factor 19 (PRPF19) was a hub gene in Turquoise that are enriched in cell cycle, spliceosome and DNA replication-repair in this study. PRPF19 is a component of Prp19 complex in spliceosome that regulates RNA splicing by participating in a post-transcriptional regulation of eukaryotic genes [77]. Prp19 complex is a highly evolutionarily conserved splicing factor, and involved in various cellular processes such as pre-mRNA splicing, ubiquitin-proteasome, cell proliferation, response to DNA damage and apoptosis as well as roles in human disease [78,79]. In addition, PRPF19 gene (505 amino acid sequences) obtained in this transcriptome study showed high identity with *Larimichthys crocea* (XP_010746969.1; 99.80%), *Lates calcarifer* (XP_018554976.1; 99.80%), and *Paralichthys olivaceus* (XP_019941218.1; 99.81%), indicating there is a high degree of homology between fish species. Expression of PRPF19 gene was increased as soon as RBIV was inoculated into rock bream blood cells, and it was up-regulated continuously during the observation period (Figure 5**F**). Although more research on the role and function of PRPF19 in fish disease is needed because there has been no study of Prp19 in fish so far, we suggest that this very sensitive gene might act as potential biomarker in rapid diagnosis of RBIV infection in fish.

## 4. Materials and Methods 

### 4.1. Ethical Statement

Animal experiments performed in this study did not involve endangered or protected species. All experiments were carried out in accordance with the guidelines and regulations. This study was approved by Ethics Committee of Pukyong National University (approval number: 2017-09) according to the Bioethics and Safety Act ministry of the Ministry of health and welfare in South Korea.

### 4.2. Sample Collection and Preparation

Light and heavy RBIV-infected rock bream (*n* = ~ 200 for each farm, body weight = 20−30 g) were obtained from two fish farms in Tongyeong and Geoje-do, respectively. Fish in both farms were originated from the same hatchery. Each fish group was acclimated in a separate tank (1000-L) in an aquarium of Pukyong National University. Mortality was monitored for three weeks. Fish were assigned into a heavy infected group designated as 0H (heavy RBIV infection) or 0MH group (mixed heavy RBIV and bacterial infection) based on viral loads in spleen (Appendix A). Mortality in the heavy infected group was high for two weeks. However, no mortality occurred in the third week. The remaining fish were sacrificed and designated as 3L group (light RBIV infection). Fish in lightly to negative infected group as a control were sampled at the same sampling time points as 0C and 3C groups. Head kidney, liver, and spleen were taken from randomly selected fish (*n* = 5) in each group. Weight of the spleen was measured to determine spleen index (SI = mg of spleen/g of body weight) [14]. Detection and quantification of RBIV were performed using nested PCR and real-time PCR with splenic DNA and primer sets targeting major capsid protein (MCP) gene of RBIV (Appendix A and Appendix A). For Iso-seq and transcriptome analysis, a piece of each tissue sample was treated with RNAlater (Qiagen, Germany) and stored at −80 °C. For ^1^H-NMR-based metabolome analysis and RBIV quantification, the remaining tissue samples were frozen in liquid nitrogen and stored at −80 °C, and detail methods were described in Supplementary Methods and Results.

### 4.3. RNA Extraction for Next Generation Sequencing (NGS)

Head kidney, liver, and spleen taken from both healthy and RBIV-infected rock bream were used to generate reference transcripts through isoform sequencing (Iso-seq^TM^) (Supplementary Method and Results). For Illumina sequencing, only spleen samples were used. Total RNAs were extracted from tissue samples stored in RNAlater using TRIzol Reagent (Invitrogen, USA) according to the manufacturer’s instructions. Concentration and integrity of total RNAs were determined using Quant-IT RiboGreen (Invitrogen, USA) and TapeStation RNA screentape (Agilent Technologies, Santa Clara, CA, USA), respectively. Only high-quality RNA preparations with RNA integrity number (RIN) greater than 7.0 were used for RNA library construction.

### 4.4. cDNA Library Construction and Illumina Sequencing

For sequencing, cDNA libraries were constructed using Illumina TruSeq RNA Sample Preparation Kit (Illumina, San Diego, CA, USA) following the manufacturer’s protocol. Briefly, RNA molecules containing poly(A) tail were purified from 1−2 ug of total RNA pool with oligo-dT magnetic beads, then fragmented mRNAs were reverse-transcribed to first strand cDNAs using random primers. Synthesized cDNA fragments were ligated with adapter sequences, amplified, and qualified following protocols provided by Illumina. They were then sequenced on a HiSeq2500 platform (Illumina, USA) for 100 bp paired-end.

### 4.5. De Novo Assembly, Functional Annotation, and Differentially Expressed Gene (DEG) Analysis

Sequence reads generated from Illumina were filtered by removing adapter sequences and trimming low-quality reads having ambiguous bases or with average length less than 20 bases. RNA-seq by Expectation Maximization (RSEM) was used to align reads with unigenes generated by Iso-seq (Supplementary Methods and Results) and estimate read abundance [80]. Fragments Per Kilo-base of exon per Million mapped fragments (FPKM) method was used to calculate expression levels of transcripts. Annotations for unigenes were performed through BLASTx with e-value cutoff of 1e-10 against Swissprot and nr database in NCBI (version 2.6.0+) [81]. InterProScan (version 5.17-56.0) was used for domain search [82]. Results with blast top hit were used for further analysis. Kyoto Encyclopedia of Genes and Genomes (KEGG) pathway analysis and annotation were conducted using BLASTx tool with database of available fish species and KEGG Automatic Annotation Server (KAAS) (http://www.genome.jp/tools/kaas) with default parameters [83]. Differentially expressed gene (DEG) analysis between 0C and other groups was performed using Tag Count Comparison (TCC) package through iterative DEGES/edgeR method based on Trimmed Mean of M-value (TMM) normalization with false discovery rate (FDR) < 0.05 in R [84].

### 4.6. Weighted Gene Co-Expression Network Analysis (WGCNA)

Co-expression analysis was performed using WGCNA package in R [85]. From 25 samples, 35,861 genes (52.57% of unigenes) over FPKM value at 0.01 with high confidence [86,87] were used for network construction. Signed co-expression similarity s_ij_ was calculated as s_ij_ = (1 + cor(x_i_, x_j_) )/ 2. A soft thresholding power β = 16 (R^2^ = 0.9070) was determined for this study. It was used to calculate adjacency matrix (a_ij_ = s_ij_^β^) and topological overlap matrix (TOM) dissimilarity. Blockwise Modules were used to construct network and define gene modules automatically with the following settings: corType = “pearson” and mergeCutHeight = 0.25. Gene significance (GS) was then defined as correlation between expression profile and viral copy to identify mechanisms of host defense against RBIV infection. Module membership (MM) was defined as correlation of eigengene in a given module and gene expression profile. Modules satisfying conditions of |GS versus MM| > 0.5 and *p*-value < 0.05 known to be highly related to viral copies were analyzed in this study. Intramodular hub genes in interesting modules were detected as highly connected genes with a high module membership and highly correlated with viral copy number. Functional annotation enrichment analysis of all eigengenes in each module was performed using BiNGO with default setting in Cytoscape (version 3.6.1) [88] for Gene Ontology (GO) analysis. KAAS and web-based ConsensusPathDB [89] (http://ConsensusPathDB.org) were used for KEGG pathway analysis. 

### 4.7. Gene Expression Analysis in Rock Bream Blood Cells Infected with RBIV

To investigate effects of RBIV on blood cells including RBCs, leukocytes, and thrombocytes of rock bream based on results of functional analysis of transcriptome, quantitative real-time PCR (qRT-PCR) was performed to determine expression of hub gene, apoptosis, pattern recognition, platelet activation, and immune-related gene (Appendix A). Blood was collected from caudal vein of each RBIV-negative fish (average BW = 330 g) and placed in a sodium heparinized tube and stirred for 30 min. Blood was suspended in L15 supplemented with 5% FBS (*v*/*v*) (Gibco) and 1% antibiotic/antimycotic solution (*v*/*v*) at a ratio of 1:4. This was placed in a consecutive density of Percoll gradient (34% and 51%) and centrifuged at 400 × *g* for 25 min at 4 °C. After removing medium and plasma components, cells were washed with L15 medium twice and adjusted to 2 × 10^8^ cells/flask in a 25 cm^2^ tissue culture flask (Corning, Corning, NY, USA). After inoculation with a 200 µL of viral suspension (1.03 × 10^9^ viral copies/100 µL in suspension) with a multiplicity of infection (m.o.i.) of 10, cells were incubated for 3, 6, 12, 24, 48, and 120 h. We observed changes in blood cells at sampling time point under microscope after incubation with RBIV. Total RNA was extracted from cells using combined protocol with TRIzol and easy-spin^TM^ Total RNA extraction kit (iNtRON biotechnology, Korea). cDNA was synthesized from 250 ng of total RNA in each sample using a PrimeScript^TM^ RT reagent kit (TAKARA Bio, Japan) according to the manufacturer’s instructions. Quantitative RT-PCR was performed in a 20 µL reaction mixture containing 10 µL of qPCR green 2× master mix kit (m.biotech, Korea), 2 µL of cDNA template, 1 µL of each primer (10 pM), and 6 µL of DEPC-treated water using Exicycler 96 Real-Time Quantitative Thermal Block (Bioneer, Korea). qPCR conditions were as follows: 95 °C for 10 s, followed by 45 cycles of 95 °C for 5 s, 58 °C for 20 s, 72 °C for 20 s. The β -actin gene was selected as an internal reference as previously described [90,91,92,93]. In particular, Zhang et al. [90] found that β-actin showed more reliable results in the spleen of rock bream infected with Megalocytivirus among various housekeeping genes. Among three primer sets for β-actin gene [17,90,91], a set used in a previous study [91] was finally selected for this study after we investigated efficiency of all the primer sets (data not shown). All samples were analyzed in triplicate. All data are expressed as mean ± standard deviation (SD) relative to the expression of β-actin gene using the 2^−ΔΔCT^ method [94].

### 4.8. Data Accessibility

Raw reads generated from PacBio and Illumina platforms were deposited into Sequence Read Archive (SRA) of NCBI with accession numbers of PRJNA511555 and PRJNA511128, respectively.

## 5. Conclusions

This is the first attempt to analyze the transcriptome in the spleen of rock bream infected with RBIV using WGCNA and integrate with NMR-based metabolome data. Based on results obtained in this study, pathways including DNA replication, transcription, translation system, aerobic glycolysis (Warburg effect), and TCA cycle to increase energy were up-regulated in heavily infected rock bream. It seems like RBIV also down-regulates signaling in or between cells and inhibits innate and adaptive immune responses such as antigen presentation, lymphocyte differentiation, and cell receptor signaling. In particular, RBIV infection seems to affect activities of RBCs, thrombocytes, and lymphocytes of rock bream. In addition, RBIV infection may decrease the integrity and cell-to-cell interactions of infected cells. It causes death of infected cells by inducing mitochondrial mediated apoptosis (Figure 6). Findings of this study provide insight into interaction between fish and RBIV, and fundamental information for the prevention of RBIV infection. This study is also very essential in terms of methodology as it shows a new direction and possibility of co-expression network analysis for fish transcriptome analysis.

## Figures and Tables

**Figure 1 ijms-21-01707-f001:**
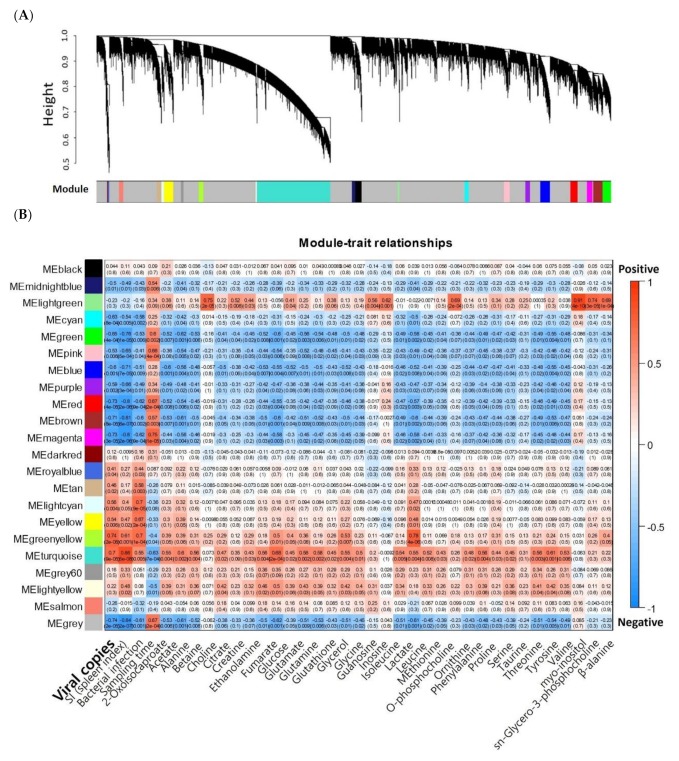
(**A**) Gene dendrogram and 22 different colors for modules obtained in this study. Dendrogram was calculated by average linkage hierarchical clustering. Module colors were determined by cutreeHybrid. (**B**) Module-trait associations. Correlations between module eigengenes (MEs, *y*-axis) and clinical traits (*x*-axis) including viral loads, spleen index (SI), and relative concentration of several metabolites detected by NMR were calculated. Each block represents both Pearson’s correlation coefficient and *p*-value (in bracket).

**Figure 2 ijms-21-01707-f002:**
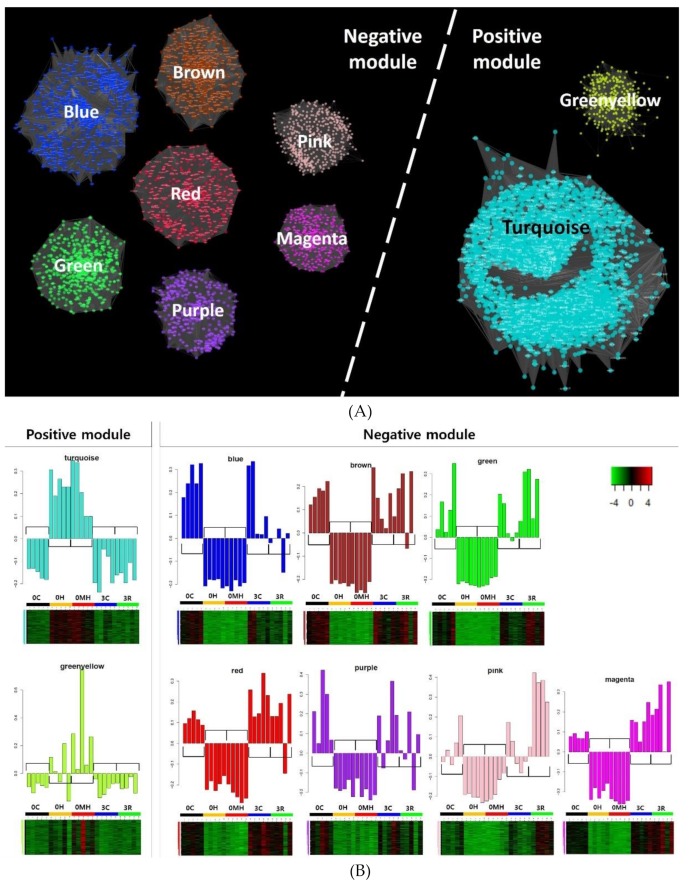
Construction of co-expression network (**A**) and bar graph and heatmap (**B**) representing eigengene expression patterns in selected nine modules (|GS vs. MM| > 0.5, *p*-value < 0.05) having positive and negative correlations with splenic viral load. In bar plots, x- and y-axis show samples and module eigengene representing the first principal component of gene expression matrix in each module, respectively. 0C; uninfected healthy fish at week 0 as control; 0H, heavy infected fish at week 0; 0MH, heavy mixed RBIV and bacterial infected fish at week 0; 3C, uninfected healthy fish at week 3; 3L, light infected fish at week 3.

**Figure 3 ijms-21-01707-f003:**
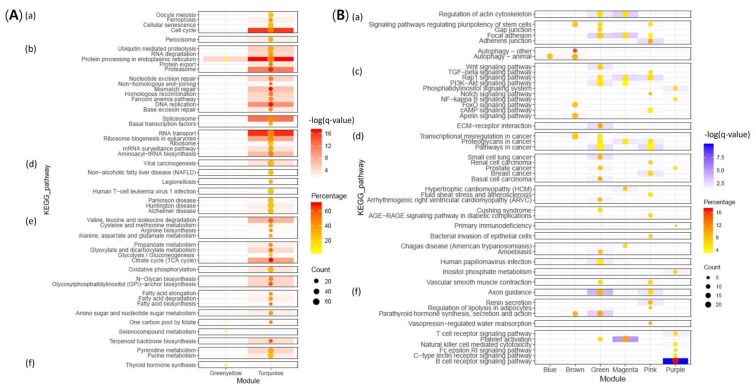
Visualization of all results from KEGG over-representation analysis of positive (**A**) and negative (**B**) modules (*q*-value < 0.05) performed by consensuspathDB. Box color represents the significance level (that is, deeper color means more significant). Circle size and color intensity in box represent the number of genes and percentage mapped in the pathway, respectively. Top categories in KEGG orthology (KO): (a) cellular processes, (b) genetic information processing c) environmental information processing, (d) human diseases, (e) metabolism, and (f) only immune systems in organismal systems.

**Figure 4 ijms-21-01707-f004:**
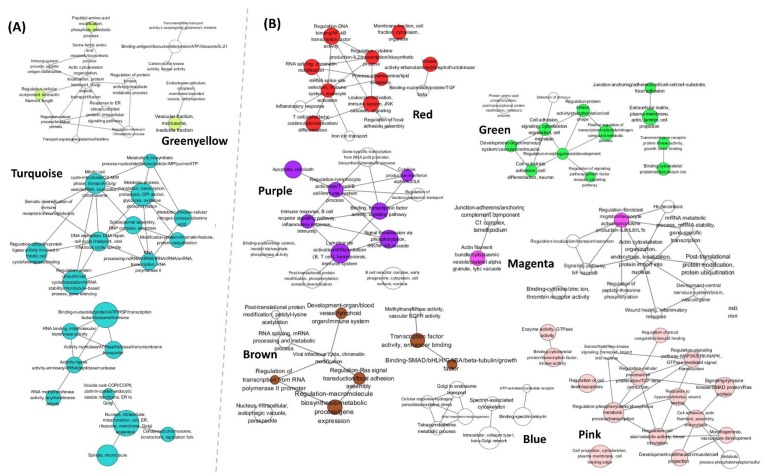
GO over-representation analysis results of positive (**A**) and negative (**B**) modules. Enriched GO terms by BiNGO (adjusted *p*-value < 0.05) were clustered by AutoAnnotate (version 1.2) with community cluster (GLay) algorithm in ClusterMaker2 (version 1.2.1) and annotated with WordCloud (version 3.1.2) in Cytoscape. Adjusted *p*-value was corrected using a Benjamini and Hochberg False Discovery Rate (FDR). Node size and color indicate the number of enriched GO terms and module color, respectively. The edge between nodes means presence of shared genes.

**Figure 5 ijms-21-01707-f005:**
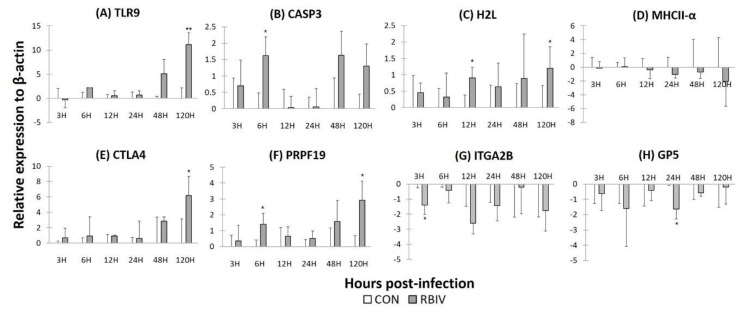
Relative expression levels of genes related to pattern recognition receptor (PRR) (**A**), apoptosis (**B**), antigen presentation (**C**,**D**), immune (**E**), hub gene, and (**F**) platelet activation (**G**,**H**) in blood cells of rock bream after RBIV inoculation. The expression level of β-actin was used as an internal control. Each experiment was performed in triplicate. Each bar and error bar represent log_2_FC value and standard deviation (SD), respectively. (*) and (**) indicate *p* < 0.05 and *p* < 0.01, respectively, relative to the control. *TLR9*, Toll-like receptor 9; *CASP3*, Caspase-3; *H2L*, H-2 class I histocompatibility antigen, L-D alpha chain; *MHC2α*, MHC class II antigen alpha chain; *CTLA4*, Cytotoxic T-lymphocyte protein 4-like; *PRPF19*, Pre-mRNA processing factor 19; *ITGA2B*, Integrin alpha 2B; *GP5*, Platelet glycoprotein V.

**Figure 6 ijms-21-01707-f006:**
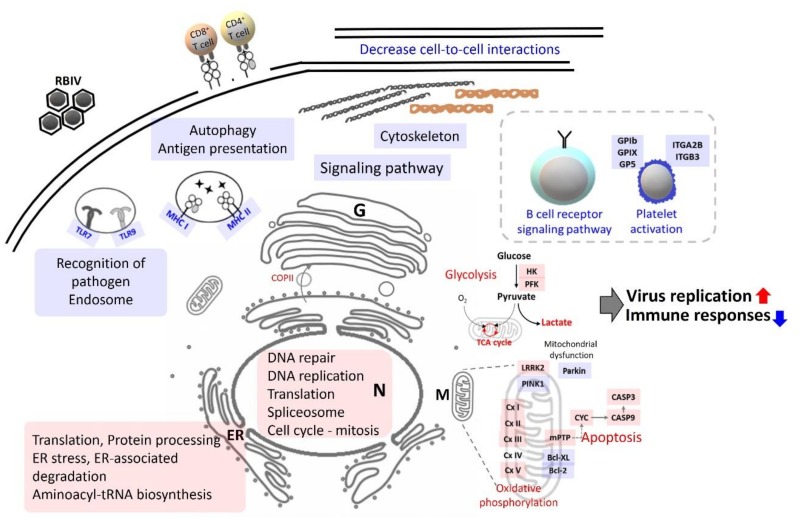
Schematic proposal showing how rock bream iridovirus (RBIV) could interact with cells in spleen of heavy-infected rock bream. Red and blue colors indicate up- and down-regulation, respectively, in host cells. N, Nucleus; ER, endoplasmic reticulum; G, Golgi apparatus; M, mitochondria; TLR, Toll-like receptor; MHC, major histocompatibility complex; Cx, Complex; COPII, coat protein II.

**Table 1 ijms-21-01707-t001:** Summary of 9 modules (absolute correlation value between gene significance (GS) and module membership (MM) greater than 0.5, and *p*-value < 0.05) and the most significant enriched KEGG pathways (*q*-value < 0.05) and GO terms (adjusted *p*-value < 0.05) in each module.

	Modules	KEGG (*q*-Value < 0.05)	GO (Adjusted *p*-Value < 0.05)
	Gene Cluster	No. of Genes	GS vs. MMCorrelation	*p*-Value
Positive	1	Turquoise	5136	0.53	< 1 × 10^−200^	Protein processing in ERCell cycle, RNA transportProteasomeSpliceosomeDNA replication	Nucleus, Spliceosome, Mitochondria, Intracellular, Vesicle (COPI/COPII), Transcription, Translation, tRNA, Metabolic process, Mitotic cell cycle, Glycolysis, DNA replication-repair, Protein ubiquitin
2	Greenyellow	312	0.74	2.50 × 10^−55^	Protein processing in ER	Amino acid modification, ER stress, Antigen binding, Vesicular fraction
Negative	3	Blue	682	−0.68	1.00 × 10^−93^	Autophagy	Spectrin-associated cytoskeleton, Golgi to endosome transport
4	Brown	665	−0.56	3.70 × 10^−56^	Autophagy	Macromolecule biosynthesis, Transcription factor activity, Gene expression
5	Green	550	−0.51	9.50 × 10^−38^	Axon guidanceFocal adhesion	Cell adhesion, Extracellular matrix
6	Red	503	−0.52	3.40 × 10^−36^	No hit	Regulation (cellular process, signaling pathway, DNA binding), Membrane fraction, RNA splicing, T cell costimulation, Leukocyte activation, Cytokine
7	Purple	325	−0.60	3.70 × 10^−33^	B cell receptor signaling pathway	Lymphocyte activation (B/T cell), B cell immune system, Binding
8	Pink	415	−0.53	2.00 × 10^−31^	Rap1 signaling pathway,Adherens junction	Development, Morphogenesis, Signaling process, Cytoskeleton, Ras protein, Apoptosis, Wound healing
9	Magenta	377	−0.53	1.10 × 10^−28^	Platelet activation, Focal adhesion, Regulation of actin cytoskeleton	Junction, C1 complex, Actin, Platelet alpha granule, Wound healing, Vesicle
		Total	8965

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
