# Peer review of "Co-Expression Network Analysis of Spleen Transcriptome in Rock Bream (Oplegnathus fasciatus) Naturally Infected with Rock Bream Iridovirus (RBIV)"

_ijms, 2020, doi:10.3390/ijms21051707_

Round 1

Reviewer 1 Report

Overall a valuable transcriptome analysis of rock bream infected with RBIV.  The identification and publication of these results will help future researchers with studies into this system.  

A reminder of the abbreviations used for the treatment groups in the text would be useful before diving into the bulk of the results.

In figure 5, B-actin was chosen as a housekeeping gene, despite Actin regulation being noted in table 1.  It was not discussed or mentioned whether or not this housekeeping gene was checked for stability between mock and infected blood cells prior for use in these experiments.  I would question whether it is a suitable normalization gene and would test other options or use a basket of potential options.

PRPF19 discussion could be improved upon to include if this gene was identified in any other RNA-seq studies of fish infected with RBIV, if not identified this should be addressed and commented upon.     

Author Response

 Thank you for your comment. 

Reviewer 2 Report

The study of Kim et al. evaluates the transcriptomic response in spleen from rock bream naturally infected with rock bream iridovirus (RBIV). The results are interesting and the discussion is correct. The title seems oversold to me: the core of the study is not based on network analysis. In fact, the networking is only based on processes but not on DEGs. Furthermore, the network is only to highlight those GO processes finding in the study in an already pre-structured (by default) network. Therefore, the network just provides a representation of the GOs in a jointly manner rather than providing new antecedents that could not be found without this strategy. Some methodological aspects need to be clarified. On the other hand, as a major concern, based on your results, I don’t see clearly the reason to carry out the experiment detail in section 4.7. The justification for the gene expression analysis in rock bream blood cells infected (or incubated?) with RBIV has a lack of robustness. Moreover, there is no detail to characterize such “blood cells” (number of cells classified by cell-type) in order to have an idea of the main cell-driven responsible for the response indicated. Another concern is related to the huge amount of less-exploited results included in this study: the majority of them are poorly explained and, in consequence, it is very poor the profit you take from them. The quality of the figures must improve (they become blurry when zoomed in to distinguish values; Figure 1B is a good example). Some concerns are detailed below by sections:

Title. “Co-expression network analysis” is an oversold concept for this study. In fact, the authors do not take profit of the GO functional network introduced in the text (Figure 4). I strongly suggest reconsider the title.

Abstract

L26-27. What is the relationship between the “lack of effective treatment strategies” and “no transcriptomic studies on RBIV-infected rock bream”? Please clarify.

L45-47. “expression of this gene using qRT-PCR was significantly increased in blood cells of rock bream immediately after viral infection”. This upregulation at early infection time, is it enough to consider this gene as a biomarker for diagnosis in the context of Aquaculture?

L51. Please mention the gene full name together with the PRPF19 acronym.

L76-77. “due to profundity of RBIV on rock bream”. I don´t get the relevance of this last phrase. I suggest delete it.

L110-111. Please include the mortality curve in the ms. The mortality progression is also relevant in the mortality curves (not only the cumulative mortality values).

L117-118. “Mean splenic viral load of five randomly selected fish 117 was 4.15 ± 4.14 × 103 copies/mg of spleen”. Please refers this information to 3L group.

L121. “clustered well”. Please rephrase.

L121. “samples in each group were clustered well”. What about the sample from the “3C group” that looks bit far compared to the rest of the well-clustered transcriptomes? This response should be included in the results.

Results.

L122. What is your interpretation for the tight relationship between the 0MH and 0H observed? It is curious because the evident difference between them is the presence of Vibrio scophthalmi and V. harveyi in the 0MH group. Why do you think the spleen transcriptome does not show the difference between both groups? Moreover considering spleen as immunological tissue. This concern has to be included in the discussion section.

L123-124. “A total of 22 modules were created by WGCNA. They were labeled by colors as shown in Figure 1(A)”.For someone not familiarized with these representations, the figure does not help to identify precisely the 22 modules you mention. Please make the corresponding rearrangements to make the figure understandable with the message. At the same time, please provide a description of the colors indicated below the dendrogram. In the state how it is now the figure and its description in result section, it is very hard to understand what is the correlation between the dendrogram and module color indicated below, and the scope of these data by itself.

L126. “relative concentration of metabolites”. The results for metabolites have not been described so far. I suggest to introduce the results for metabolites first. Otherwise, the written becomes confusing for the reader.

L127. Please provide definitions for GS and MM (the first time you mention these acronyms in the text).

L129. “were confirmed were confirmed to be up- and down-regulated in positive and negative modules, respectively” What are those positive and negative modules? What are the criteria to classify them as “positive” or “negative”? Please provide such info.

L159. “in Turquoise module”. Please homogenize throughout the text.

L164. Please provide a definition for the gene acronyms.

L168. “In RNA transport pathway…” please provide the number of genes for each path detailed.

L178-181. How many genes are present in each described process? Please include this info in the ms.

L185. “All genes”. How many? Please include this information.

L194-195. Please detail the number of genes for each pathway.

L241-242. “We hypothesized that RBIV would interact with blood cells of rock bream based on our 241 transcriptome results”. This hypothesis is very poor. A better explanation of the reasons that justify this analysis must be indicated in the text. Based on your results, the analysis of interesting specific immune-related cell types would be interesting.

-Why did you choose to evaluate the effect of RBIV in blood cells?

-What does it mean blood cells? Did you check the majority population present in your sample?

L242. Linked with the last comment, there is no table 2 in the text. Thus, the reason to justify the gene expression analysis of the genes indicated in Figure 5 is missing. Please include all these concerns in the text.

Discussion.

L282-283. How is the modulation of these processes in your study? Please provide this information.

L290-295. Do the viruses mention any relationship with RBIV that justify the association you are proposing? Please justify in the text.

L296. According to my last comment, there is any relationship between these viruses and RBIV? Please justify in the text. Please make the same exercise for all the viruses mentioned in the discussion section.

L304. “ER” meaning?.

L309. Please provide definitions for the acronyms used. Make the same for all the acronyms/abbreviations the first time you mention them throughout the text.

L346-347- “Thrombocyte…”. Lack of sense. Please rephrase.

L355-358. The time indicated (3 hpi, 1 dpi), is it enough to provoke thrombocytopenia in RBIV-infected fish? How long is the RBIV life cycle? Please justify.

Conclusions.

L518. From my point of view, from the results obtained in this study it is not possible to suggest the virus use replication, transcription and translation host-cell systems. The modulation of these mechanisms could be an intrinsic mechanism of response against the viral infection. So, I don´t see how the authors can support this statement.

Methodology.

L422. Please detail the guidelines and regulations to which you refer.

L453. “Then,” instead “after”.

L491. “infected” or “incubated with RBIV”? The second one seems more adequate to me.

L496-500. With the protocol indicate, what cell type was the idea isolate? What are the antecedents that support such decision? What cell type(s) did you isolate finally? Did you check the majority population present in your sample? Did you characterize the isolated cells in your sample? What methods did you use for? “blood cells” is too general. Please precise.

L495. Does “healthy rock bream” mean “RBIV negative fish”? Please clarify.

L501. What is the justification for the use of such viral copies (10^9 copies/100 ul suspension) for incubation? None of the fish evaluate had such magnitude order (maximum 10^8 copies/mg). Please justify and provide details of your decision in the body text.

L501. Did you effectively use 200 ul of viral suspension in a concentration of 10^9 copies/100 ul in suspension? Please clarify.

L502. Please indicate all the values on the same time scale (i.e. hours). Accordingly, make the replacements throughout the text.

L511. In regard to the use of beta-actin as a single reference gene, the use of these reference genes is not always good references and as such needs to be validated for tissue in every experiment. How stable was the reference gene and what was the reference gene chosen in this study? Was there one or multiple as has been requested/required by the MIQE guidelines (Bustin et al., 2009). In following with these guidelines, efficiencies of all genes (i.e. GOIs and refs) should be given in the primer table along with primer seqs, product sizes and annealing temps, among other relevant aspects.

Figures.

Figure 1A. Please include in the image a color scheme legend to represent the 22 modules represented with color (as indicated in caption).

Figure 2:

-Please provide a “y-axis” definition.

- Please provide a color gradient scheme as explanation of heatmap representation included in the figure.

-The color bars behind each named group, does it mean something? Please clarify/modify accordingly.

Figure 3:

-From top to the bottom, the order of –log(q value), percentage, and count is not the same between “Figure 3A” and “Figure 3B”. Please homogenize.

-Please use a graphic design resource for clearly indicate in the figure the limit for the KO categories already indicated.

-What is your explanation for the presence of KO category associated to human disease? How this information help to unveil the transcriptomic response of rock bream to RBIV?

Figure 5. What is the reason you evaluate the expression for TLR9 instead of TLR7? From a viral infection perspective, TLR7 may look a more interesting gene than TLR9.

Figure 6. A better representation for a “decrease cell-to-cell interaction” is recommended.

Tables.

Table 1:

-“Summary of the 9 modules with significant correlation with splenic viral load (absolute…”

-Please provide a definition for the abbreviations GS and MM. Table captions must be understandable by itself.

Table S1. Please provide:

- a title for all the columns would be grateful (i.e. “per fish”) for “viral copies…” and “spleen index”.

- For the spleen index, please also provide the mg of spleen and the g of body weight.

- What is the reason to get an ND result for viral copies in the 3C group? It is curious considering you have very low copies (i.e. 1.26x10^1) in the 0C group.

- How did you calculate the viral copies? Please detail the formula used. This information is very important because some common omissions of variables in the formula provoke mistakes in its estimation.

- In regard to the viral copies estimation, what is the minimum and maximum estimation limit? Please include this information in the methodology section at body text.

Author Response

 Thank you for your comment.

Round 2

Reviewer 2 Report

The authors have asked/include most of the concerns/recommendations suggested. Nonetheless, in regard to qPCR analysis, there is no extra information clarifying the use of more than one reference gene. According to MIQE guidelines (Bustin et al., 2009), this information must be included in the ms. This suggestion was already included in the first round of revision. On the other hand, it is a petty the authors did not establish a more strong relationship between the ms and the supplementary data provided in order to certainly justify the huge amount of data attached to this ms. Despite the foregoing, it is grateful to the authors for the immense amount of supplementary information included in the ms.

A couple of minor comments are also detailed below:

L3. “pre-mRNA processing factor 19 ( PRPF19)”

L113-114. “Mortality in heavily infected group was 36.5, 55.5%, 113 and 0% at week 1, 2, and 3, respectively”. Regarding the percentages, are values based on the total population at the beginning of the measure or are the percentage of survival fish (excluding losses) at each week? Please clarify in the ms.

qPCR analysis. As I mentioned in the first round of revision, the use of these reference genes are not always good references and as such needs to be validated for tissue in every experiment. Accordingly, the justification of beta-actin as an internal control based such gene is the most commonly used gene in similar previous studies using rock bream is clearly not enough. The beta-actin as reference gene has to be validated as the best possible based on multiple reference gene analyses according to the MIQE guidelines (Bustin et al., 2009). In following with these guidelines, efficiencies of all genes (i.e. GOIs and refs) should be given in the primer table along with primer seqs, and annealing temps, among other relevants aspects. These data for the primers are not present in the revised version. Please include them.
